# A Burn-in Test Station for the ATLAS Phase-II Tile-Calorimeter Low-Voltage Power Supply Transformer-Coupled Buck Converters

Ryan Peter Mckenzie  on behalf of the ATLAS Tile Calorimeter System

School of Physics and Institute for Collider Particle Physics, University of the Witwatersrand, Johannesburg 2050, South Africa; ryan.peter.mckenzie@cern.ch

**Abstract:** The upgrade of the ATLAS hadronic tile-calorimeter (TileCal) Low-Voltage Power Supply (LVPS) falls under the high-luminosity LHC upgrade project. This article serves to provide an overview of the development of a burn-in test station for a Phase-II upgrade LVPS component known as a Brick. These Bricks are radiation hard transformer-coupled buck converters that function to step-down bulk 200 V DC power to the 10 V DC power required by the on-detector electronics. To ensure the high reliability of the Bricks, once installed within the TileCal, a burn-in test station has been designed and built. The Burn-in procedure subjects the Bricks to sub-optimal operating conditions that function to accelerate their aging as well as to stimulate failure mechanisms. This results in elements of the Brick that would fail prematurely within the TileCal failing within the burn-in station or to experience performance degradation that can be detected by followup testing effectively screening out the 'weak' sub-population. The burn-in station is of a fully custom design in both its hardware and software. The development of the test station will be explored and the preliminary burn-in procedure to be employed will be presented. The commissioning of the burn-in station will be presented along with a summary and outlook of the project.

**Keywords:** ATLAS; TileCal; Phase-II upgrade; quality assurance testing; burn-in; transformer-coupled buck converters



## 1. Introduction

The Large Hadron Collider (LHC), located at CERN, is a two-ring-superconducting-hadron accelerator and collider [1]. The LHC is the latest addition to CERN's accelerator complex and is the last machine in a chain of successive particle accelerators that function as pre-accelerators for the LHC. Its goal is to study particle physics processes at energies and luminosities that have not been reached before. The LHC was successfully commissioned in 2010 for proton–proton collisions with a centre-of-mass energy of 7 TeV. The center-of-mass energy of the LHC progressively increased to 13 TeV during the data taking period between 2015 and 2018 with a value of 13.6 TeV having been recorded in July of 2022.

Two counter rotating particle beams are accelerated, focused, and then collided at the four distinct interaction points located along the LHC's circumference. Each of these points have an associated detector (experiment) that is responsible for recording the resulting collisions. The ATLAS (A Toroidal LHC ApparatuS) general-purpose detector is located at point-1 on its circumference [2]. The detector has been successfully collecting data since November 2009 and is currently engaged in RUN-3 data collection that will take place until 2026 when the Long-shutdown-3, associated with the Phase-II upgrade, will begin.

ATLAS is itself a composition of multiple sub-detectors each of which is specialized to detect specific types of particles. This, as different kinds of particles interact with detector materials in different ways. The particles can be distinguished based on the unique signal that they create in these detectors. The sub-detectors of ATLAS can be divided into

three categories: tracking detectors, which are closest to the beam pipe, followed by the calorimeters and the muon system, which covers the outermost part of the detector.

The TileCal is a sampling calorimeter that forms the central barrel region of the Hadronic calorimeter of the ATLAS experiment at the Large Hadron Collider (LHC). It performs several critical functions within ATLAS such as the identification of hadronic jets and measurement of their energy and direction. It also participates in the measurement of the missing transverse momentum carried by non-interacting particles, muon identification, and provides inputs to the Level-1 calorimeter trigger system. The detector is located in the pseudorapidity region $|\eta| < 1.7$ (ATLAS makes use of a right-handed coordinate system with its origin located at the nominal interaction point (IP) in the centre of the detector and the z-axis along the beam pipe. The x-axis points from the IP to the centre of the LHC ring, and the y-axis points upwards. Cylindrical coordinates (r,$\phi$) are used in the transverse plane, where $\phi$ is defined as the azimuthal angle around the z-axis. The pseudorapidity is defined in terms of the polar angle $\theta$ as $\eta = -ln \tan (\theta/2)$.) and is segmented into 3 barrel regions. The Long-Barrel (LB) region is centrally located with the Extended Barrel (EB) regions located on opposing sides of the long-barrel as see in Figure 1. Each barrel region consists of 64 wedge-shaped modules which cover $\triangle \phi \sim 0.1$. The on-detector electronics are housed within drawers located on the outer radii of the Tilecal and receive low-voltage power from Low-Voltage Power Supplies (LVPS) located in adjacent shielded boxes referred to as Fingers. The LB Modules are serviced by 2 LVPS while the EB modules are serviced by only 1 adding to a total of 256 LVPS within the Tilecal. Each LVPS contains eight transformer-coupled buck converters (Bricks) as seen in Figure 2, a fuse-board, an Embedded-Local Monitoring Board (ELMB) motherboard, a wiring harness, and a water-cooled heat sink to which the Bricks are affixed.

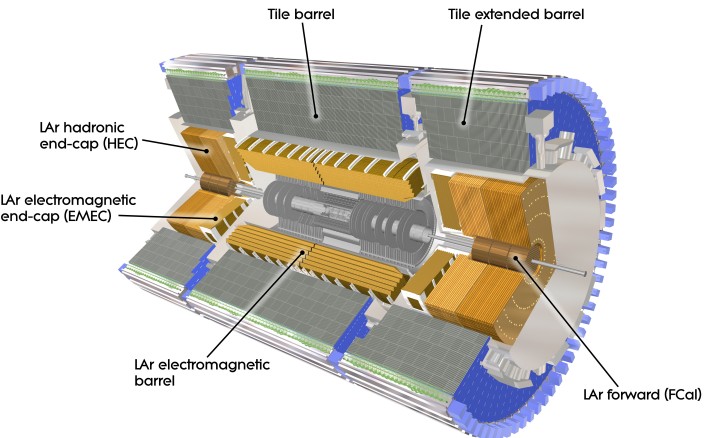

**Figure 1.** Computer generated image of the ATLAS calorimeter [3]. The LVPS Fingers are illustrated as blue boxes located on the outer radii of the ATLAS inner barrel.

In the first quarter of 2029, the start of the operation of the High Luminosity LHC is planned [4]. The resulting environment has necessitated the development of new electronics, both on- and off-detector, to ensure the continued peak performance of the Tilecal. The development, production, installation and commissioning of the new electronics falls under the Tilecal Phase-II upgrade [5–7]. The Tilecal Low-Voltage Power Supplies (LVPS) and the components therein will be upgraded as part of Phase-II upgrade. This endeavour is being undertaken jointly by the University of the Witwatersrand (Wits) and the University of Texas at Arlington (UTA) with both institutions assuming the responsibility of their local production and quality assurance testing. The number of required Bricks is equally divided between both institutions. The production of the LVPS Bricks, as with all Phase-II upgrade electronics, is subdivided into pre-production and main-production. Pre-production requires the production of 10 % of the total required Bricks prior, and in addition, to main-production. It serves as a trial production that is used to both prepare for

the main-production as well as produce spare Bricks that can be used as replacements for future failed Bricks. Pre-production is anticipated to commence at the beginning of 2023 at both institutes.

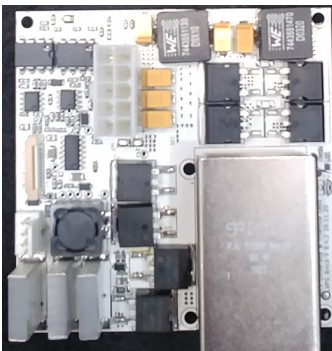 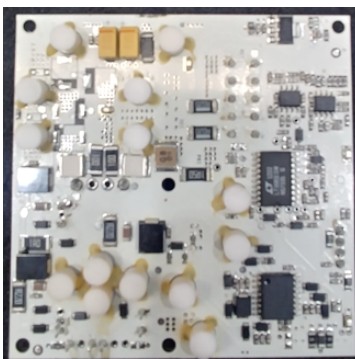

**Figure 2.** A top-view image of a V8.4.2 LVPS Brick prototype (**Left**). A bottom view image of a V8.4.2 LVPS Brick (**Right**).

Access to the LVPS Bricks is of the order of once per year, during the annual year-end technical stop, due to their location within the detector. Therefore, any Bricks that experience a failure (A failure is defined as the permanent inability of the LVPS to provide power to its associated front-end electronics, therefore requiring replacement.) will result in a portion of the module to which they provide power being offline for a commensurate period of time with the inability to collect collision data. Due to this, the reliability of the Bricks is of utmost importance. A reliability study has been conducted and a robust assurance procedure as illustrated in Figure 3 has been developed which is applied to every Brick post-production. Burn-in testing and its associated apparatus forms the crux of this article and will be covered in detail from Section 3 onward.

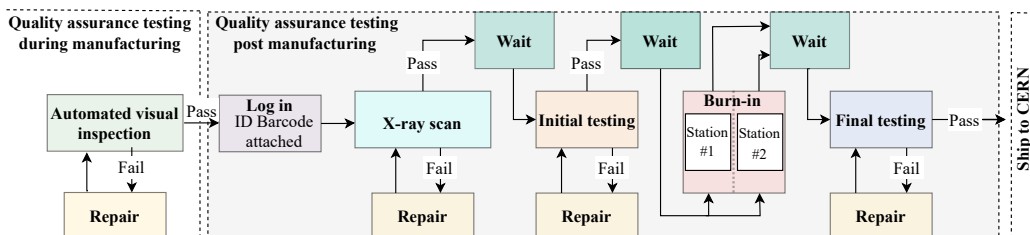

**Figure 3.** Functional block diagram of the LVPS Brick quality assurance procedure.

## 2. Phase-II Upgrade LVPS Brick

The LVPS Bricks are of an iterative design with the V6.5.4 being the first version to be installed in the TileCal in 2007. Although these Bricks generally functioned well they began to exhibit a sensitivity to trip at a rate that scaled with the luminosity of the beam. To address this issue as well as implement design changes motivated by experience with the V6.5.4 Bricks, the V7 Bricks were designed. The V7.5.0 Bricks were installed within the TileCal in 2013 [8]. The V7.5.0 Bricks are to remain within the TileCal up until the Phase-II upgrade, at which point they will be replaced with the finalized V8.5.0 Bricks. The V7.5.0 Bricks have operated reliably with an average of 2 hard failures (requiring replacement) per year over a period of 3 years since their installation in 2013. Data collection on LVPS brick failures ceased in 2016 due to the limited number of failures.

The replacement of the V7.5.0 Bricks during the Phase-II upgrade was motivated due to several key factors such as the increased radiation hardness requirements of active components necessitated by the high luminosity environment, the introduction of a third stage to the low-voltage power distribution system of the TileCal, thermal performance optimization, and the introduction of tri-state functionality. The newly introduced tri-state functionality refers to the on/off control of an individual Brick utilizing a tri-state voltage

signal. This differs from the current system which can only control an entire LVPS. The tri-state signal is so named as it can be set to one of three voltages namely the start, run and disable voltages. These voltages are 15 V, floating, and 0 V, respectively.

The radiation hardness testing, although a primary motivator for the Phase-II LVPS Bricks, is the subject of additional research and will therefore not be discussed in this paper in detail. This is justified as the radiation certification procedure of a Brick component is independent of the Burn-in procedure with few overlapping concerns such as the maximum allowable operating temperature of the certified components. Radiation hardness certification is required for all active components of the Brick. The specific radiation tests cover Total Ionizing Dose (TID) tolerance, Single Event Effect (SEE) tolerance, and Non-Ionizing Energy Losses (NIEL) tolerance. The parameters for these tests are defined by simulation of the environment within the Tilecal as a result of HL-LHC collisions.

The Tilecal Phase-II low-voltage power architecture makes use of a 3-stage system as opposed to the legacy 2-stage system. The first two stages of the new architecture are identical to those of the legacy system. These are the off-detector 200 V DC power supplies that supply bulk power to the LVPSs. The 256 LVPSs constitute the second stage. Although the second stage is still comprised of the Tilecal LVPSs in the Phase-II upgrade this is where the architecture fundamentally changes. The legacy system utilizes eight different Brick types each of which have a unique output voltage. This was required as each Brick directly powered a specific piece of the front-end electronics. The new system reduces the number of Brick types to one by the introduction of Point-Of-Load (POL) regulators. The POL regulators reside on the front-end electronics and facilitate the final stepping-down of the voltage to the values required by the local circuitry.

The thermal optimization of the Phase-II upgrade Bricks was motivated by the well understood phenomenon that an increased operating temperature of an electronic component such as a Field Effect Transformer (FET) is correlated with a reduction in operational lifetime. The thermal optimization of the LVPS Brick focused on addressing localized hotspots. The primary hotspots of concern were a result of the STB57N65M5 MOSFETs (labelled B in Figure 4) as well as the XAL1010-472MEB inductor (labelled E in Figure 4) [9]. The underlying reasons for each of the component's high operating temperature were high-switching losses and poor contact with the thermal post due to case design, respectively. Replacement components were investigated with the IRFS9N60A MOSFET and 7443551470 Inductor being selected [10,11]. It is worth noting that the IRFS9N60A MOSFET underwent radiation certification prior to being approved for use and that due to its reduced switching losses the Brick efficiency increased. Another benefit of thermal optimization of the Bricks is realized as the same active cooling system is used to cool both the Bricks as well as the on-detector electronics within the Tilecal module. The total cooling capacity of the system is also fixed and therefore any improvement in the thermal performance of the Bricks allows for the excess capacity to be utilized by the on-detector electronics as well as provide headroom.

The Phase-II upgrade Brick, of which there will be 2048 installed within TileCal, provides a nominal output current of 2.3 A at 10 V DC. At the centre of its design is the LT1681 controller chip as seen in Figure 5 [12]. It is a pulse width modulator that operates at a fundamental frequency of 300 kHz. The pulse width is controlled via two inputs, the first of which is a slow feedback path that monitors the feedback voltage with a bandwidth of approximately 1 kHz. The second input is a fast feedback path that monitors the current through the low-side transistor on the primary side. The LT1681 provides an output clock to the FET driver, which perform the switching on the primary side. The design utilizes synchronous switching, that is, both the high-side and low-side transistors turn on and conduct for the duration that the output clock is in the high state, and both are off when the clock is in the low state. When the FETs conduct, current flows through the primary windings of the transformer, which then transfers energy to the secondary windings. A buck converter is implemented on the secondary side of the transformer. The output side also contains an additional inductor-capacitor stage for the filtering of

noise. Voltage feedback for controlling the output voltage is provided. The V8.5.0 Brick utilizes the same protection circuitry implemented on previous iterations of the Brick. The purpose of this circuitry is to initiate a trip of the Brick if operating parameters exceed a specified range from nominal. The design utilizes three types of inbuilt protection circuitry, Over-Voltage Protection, Over-Current Protection (OCP), and Over-Temperature Protection (OTP). These circuits, if activated, initiated an immediate shutdown of the Brick. Their activation depends on the thresholds provided in Table 1.

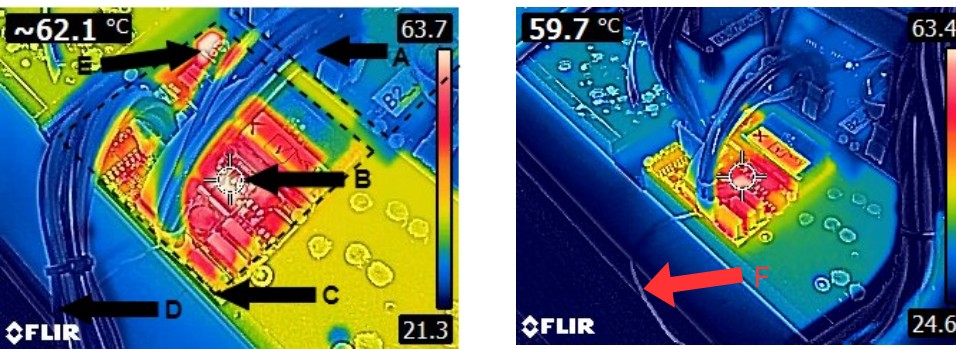

**Figure 4.** Thermal image of a V8.4.2 LVPS Brick undergoing burn-in (**Left**). Thermal image of a V8.4.2 LVPS Brick undergoing burn-in highlighting the use of an external thermocouple (**Right**).

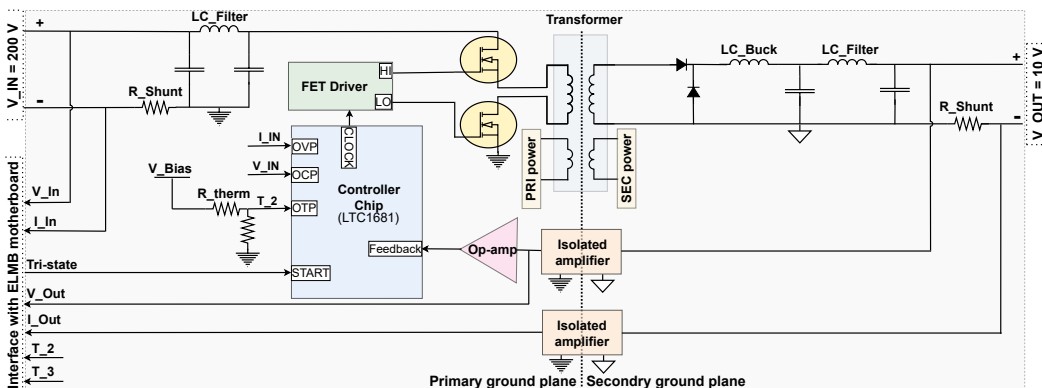

**Figure 5.** Functional block diagram of an LVPS Brick.

**Table 1.** Preliminary Phase-II upgrade Brick burn-in parameters, nominal Brick operating parameters and relevant protection circuitry trip points.

| Parameter | Burn-in | Nominal | V8.5.0 Protection Circuitry Trip Points |
|---|---|---|---|
| Operating temperature | 60 °C | 35 °C | 70 °C |
| Load | 5 A | 2.3 A | 6.9 A |
| Run-time | 8 h | - | - |

## 3. Burn-in Procedure as Part of Quality Assurance Testing

The reliability of a manufactured electronic device, such as a Brick, can differ from its predicted design reliability, at both the component and system levels. To address this phenomenon quality assurance testing is to be undertaken on all LVPS Bricks post-manufacturing [13]. The finalized sequential quality assurance procedure, as illustrated in Figure 3, is partitioned into five distinct tests. These are namely an automated visual inspection, an X-ray scan, initial testing, burn-in and final testing. Each test plays a specific role in ensuring the high reliability of the Brick population.

An automated visual inspection is undertaken by the manufacturer during the production of the Bricks. The inspection serves to identify misaligned and incorrectly populated components. Component misalignment can be exhibited to a greater degree during "small

batch productions" such as the LVPS Bricks due to a sub-optimal reflow oven temperature profile for example.

An X-ray scan of both the top and bottom side of every Brick, as seen in Figure 6, is conducted by the manufacturer. The X-ray scan serves two purposes, the first of which is to identify any short connections resulting from the reflow process. The second purpose is to assess the quality of the thermal interface between the ceramic posts and the components that are affixed to them. The eight ceramic posts located within the red boxes in Figure 6 have have their upper face metallized allowing for the components to be directly soldered to them. An undesirable yet unavoidable result of the soldering process, known as outgassing, can produce voids at the thermal interface. The X-ray scan is utilized to assess the degree of outgassing that has occurred.

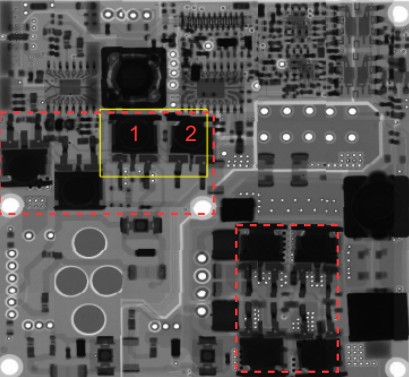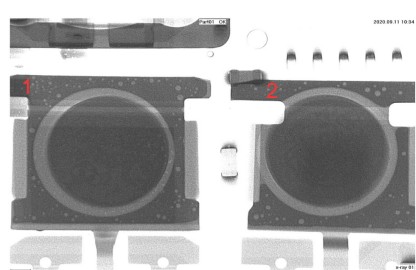

**Figure 6.** A top view X-ray scan of an LVPS Brick post reflow and before the manual population of the remaining components (**Left**). An enlarged bottom view X-ray scan of the switching MOSFET (1) and diode (2) post reflow (**Right**). These are the very same components labelled 1 and 2 in (**Left**).

Initial and final testing utilize the same test apparatus and procedure. They are so named due to their occurrence pre or post burn-in, respectively [14]. Both tests evaluate various Brick performance metrics such as output voltage, output current and operating temperature. These serve to ensure that the Brick under test is operating within its design specifications. Performance testing both before and after burn-in provides valuable insight into any performance degradation resulting from the Bricks undergoing burn-in while also serving to screen out any Bricks that may have been permanently damaged.

System level burn-in of the entire manufactured Brick population is to be undertaken. This as some patent and/or latent defects may not have been detected up to this point. Burn-in is primarily focused on detecting patent defects which appear during the early life of the Bricks but it should be noted that latent defects, that usually appear during normal operation, can be converted into patent defects via external overstress. Burn-in testing involves subjecting the Bricks to a burn-in procedure in which they are exposed to sub-optimal operating conditions such as increased operating temperature and applied load. These adverse operating conditions function to stimulate failure mechanisms within the 'weak' Brick population. This Brick sub-population is so named due to their propensity to exhibit failures or performance degradation. These Bricks need not experience a catastrophic failure during burn-in but should be easily identifiable during final testing. It is worth emphasizing that burn-in does not improve the reliability of an individual Brick but rather the reliability of the surviving brick population as any identifiable 'weak' bricks are removed from the population. If their weakness can be identified they will be repaired and subjected to the quality assurance procedure again. This process will repeat until the brick passes the entire quality assurance procedure or is deemed irreparable. A custom burn-in apparatus is required to facilitate the burn-in procedure. The development of this apparatus is described below.

## 4. Burn-in Test Station

The burn-in station is of an iterative design similarly to that of the eight Bricks that it is tasked with applying a burn-in cycle to. Each institution is responsible for the production and certification of two burn-in stations that will be utilized during the their quality assurance procedure post manufacturing. The key design requirement for the latest burn-in station is that it is able to consistently apply the burn-in procedure. This requirement is coupled with the necessity that its operation is simple and that it is safe for non-experts to operate.

The burn-in station can be decomposed into four distinct elements, namely the test-bed, the temperature control system, hardware, and software. The test-bed primarily functions to contain the majority of the burn-in station hardware and the Bricks undergoing the burn-in procedure. The test-bed is fully enclosed reducing the impact of the outside environment on the operating temperature of the Bricks and makes use of thermal/electrical insulation to prevent internal water condensation as well as potential electrical shorts while also being physically grounded.

A temperature control system is implemented to control the operating temperature of the Bricks, a key burn-in parameter, during burn-in as well as cool the Dummy-loads to which they provide power. The cooling system utilizes a commercial external water chiller that maintains the Bricks at the desired temperature set point. This temperature can be increased or decreased allowing for the Brick burn-in operating temperature to be set. The burn-in station hardware and software is discussed in Sections 4.1 and 4.2, respectively.

### 4.1. Hardware

As depicted in Figure 7, the burn-in station hardware is composed of a Personal Computer (PC), a PVS60085MR 200 V DC power supply that provided input power to the Bricks, various custom Printed Circuit Boards (PCBs), electronic components, connectors, wiring, cooling plates and a mechanical chassis presented as the test-bed [15]. The PCBs are subdivided into four types, the Main Board (MB), the Brick Interface Board (BIB), the Load Interface Board (LIB) and the Dummy Load Board (DLB). These PCBs receive 220 V AC mains power which is then rectified using DC-DC converters to supply power to the active components mounted on the PCBs. There is one MB per burn-in station responsible for addressing and demultiplexing of the interface boards. This allows a Burn-in LabVIEW Application (BLA) running on a PC to communicate via Universal Serial Bus (USB) to a Universal Asynchronous Receiver-Transmitter (UART) interface from the Programmable Interface Controller (PIC) of the MB to each PIC of the interface boards. A program on the PIC of the MB sequentially polls each of the interface boards to perform a particular task. There are eight BIBs per Burn-in station with one associated with each of the 8 Bricks undergoing the burn-in procedure. These BIBs provide control and monitoring of their respective Bricks while the LIBs provide control and monitoring of an individual dummy load. A single Brick is connected to a BIB for readout of measurements where the Brick output voltage is connected to a DLB that incorporates closed-loop Voltage Controlled Current Sink (VCCS) circuitry. The function of the DLBs is to emulate and then surpass the load that the front-end electronics would place on an LVPS Brick during operation. The DLBs make use of 4 VCCS that use high precision op-amps and N-channel MOSFETs (IRFP260NPBF) that are affixed to the Cooling Plates (CP) to dissipate the heat generated.

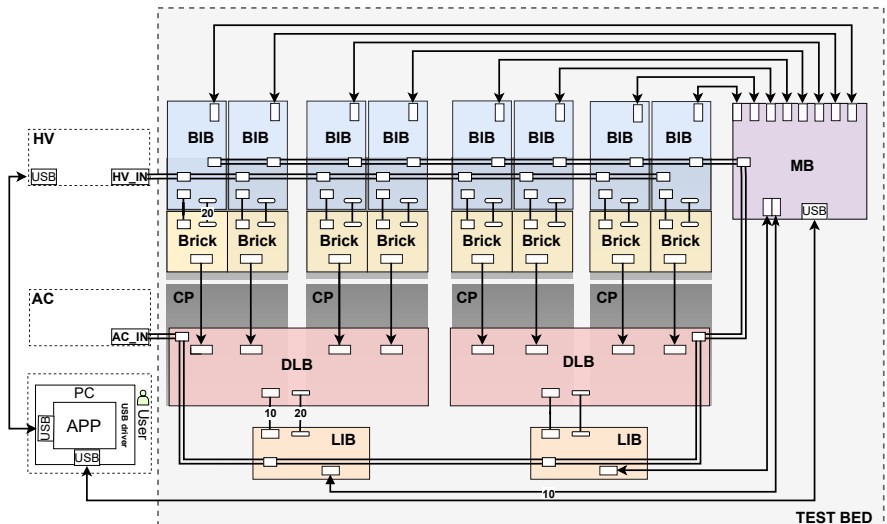

**Figure 7.** Functional block diagram illustrating the burn-in station hardware.

*4.2. Software*

The Burn-in LabVIEW Application (BLA) and PIC firmware were originally developed by Argonne National Laboratory in 2006 for the version 7 Bricks [16]. During testing and development of the burn-in station, the BLA has undergone major revisions in terms of improving code readability while only minor changes were made to the PVS60085MR power supply instrumentation drivers and the ADC channel configurations of the Interface Boards (IBs).

## 5. Preliminary Burn-in Procedure

The Burn-in procedure subjects the Bricks to sub-optimal operating conditions that function to stimulate failure mechanisms within the Bricks and is the subject of ongoing research. These conditions need to fall within the extrema allowed for by the Brick design and operation. There are two reasons for this. Firstly, the failure mechanisms stimulated must be of the kind that can be experienced during operation within the TileCal, such as an increased operating temperature, to enable effective burn-in. The second reason for limiting the burn-in parameters provided in Table 1 to be within the Bricks final operating extrema is due to the Bricks inbuilt protection circuitry. A Brick will initiate a trip if the OCP or OTP trip parameters are met during burn-in. The run-time of the burn-in procedure is an equally important parameter. One needs to maximize the efficacy of the burn-in process while maintaining a practical time frame for the burn-in of the entire Brick population. The above points combined with previous experience obtained through the burn-in of the V7.5.0 Bricks were considered resulting in the legacy burn-in parameters provided in Table 1 being selected as the starting parameters. These parameters are currently the focus of ongoing research.

## 6. Burn-in Station Commissioning

The commissioning of the first burn-in station at Wits is a vital step towards preparation for pre-production as it serves to validate the station's performance under its final operating conditions and in doing so prove its ability to undertake the role for which it was designed. The commissioning of the burn-in station utilized only one V8.4.2 Brick that was tested in each of the eight possible channels. The reason for this was entirely due to the current status of the project with the majority of the V8.4.2 Bricks having been shipped for use in test-beam campaigns at CERN. As the final V8.6 brick is expected to begin pre-production at the beginning of 2023, the decision was made to not produce another small batch of the now obsolete Brick version with the understanding that the first eight to sixteen V8.6 pre-production Bricks will be used to corroborate our findings.

A Brick (labelled C) undergoing burn-in can be observed in Figure 4 (Left). The 200 V DC input via a wiring harness (labelled D) and its output 10 V DC is supplied to its respective dummy-load channel via a similar, albeit shorter, wiring harness (labelled A). The burn-in temperature parameter of 60 °C is being met with the observed hot spot (labelled B) resulting from the primary-side switching MOSFETs.

There are distinct temperatures that need to be understood when evaluating the performance of the burn-in station as well as the thermal performance of the Brick and its components during burn-in. These temperatures differ in both their physical origin as well as the way in which they are measured. The image in Figure 4, obtained using thermography, is a useful data source that provides a global understanding of the thermal behaviour of the brick while also allowing for a focused measurement within the image cross-hairs. The infrared radiation measured and interpreted by the thermal camera originates from the surface of the Brick/components and as such does not give direct temperature information of the MOSFET junction temperature, for example. This is not problematic as in the case of these MOSFET's, and for many other components in general, there is an equation governing the relationship between the case and junction temperatures. This is particularly important as the maximum junction temperature should never be exceeded during burn-in as permanent damage can result. Thermal imaging measurements of the brick were corroborated by the use of an external thermocouple, labelled F in Figure 4 (right), with insignificant deviation observed. The use of external temperature measurement during the commissioning stage is required as the brick thermistor, the value of which is monitored by the burn-in station, is highly localized and is influenced by the thermal behaviour of the entire brick. As a consequence a thermistor value of 60 °C does not imply that the entire brick is operating at this temperature but rather that the region surrounding the thermistor is at this temperature. Therefore, the thermal distribution in relation to the thermistor temperature value needs to be understood. The thermistor utilized for the on-brick measurements is located adjacent to these MOSFETS and is also utilized in the Bricks OTP circuitry. The other observable hotspot (labelled E) originates from an inductor located on the secondary side of the Bricks. The underlying reasons for these hotspots were discussed in Section 2. The temperature monitored during the eight hour burn-in procedure is illustrated in Figure 8. The burn-in temperature was stable with a mean value of 60.7 °C and a standard deviation of $\sigma = 0.2$ °C.

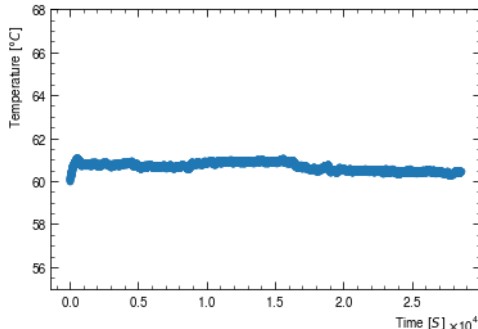

**Figure 8.** Temperature stability plot of a V8.4.2 Brick undergoing burn-in testing.

The commissioning process does not only focus on the operation/performance of the burn-in station with regards to applying the burn-in procedure to a Brick (Brick centric) but also encompasses the performance of the station's hardware and software (station centric). The burn-in station experiences thermal stress as a consequence of its operation as seen in Figure 9. The two dummy-loads, labelled G in Figure 9 (Left), are subjected to the harshest operating conditions as a single dummy-load is responsible for converting the power received from up to four bricks into heat that is to be dissipated. This can be seen in Figure 9 (Right) where a VCCS MOSFET (labelled H) is seen to be operating at 46.8 °C. This value is misleading as it is the temperature of a thermostat case that is located above and connected to the MOSFET. The thermostat was introduced as a safety feature to prevent

thermal runaway and closes when it is exposed to a temperature of 130 °C. Upon closing the thermostat grounds the MOSFET gate pin thereby temporarily disabling the VCCS. The operating temperature of the MOSFET is far higher than that illustrated in Figure 9 (Right). The temperature of the MOSFET case was measured using an external thermocouple and was found to be approximately 100 °C during nominal operation. Based on 50 W passing through the junction and utilizing a thermal resistance junction-to-case of 0.50 °C/W (as per the specification sheet) the junction temperature is 125 °C which is well below the maximum allowable junction temperature of 175 °C [17].

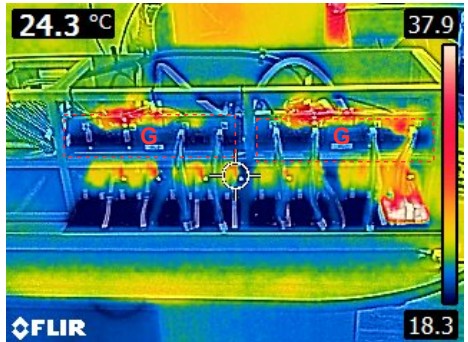 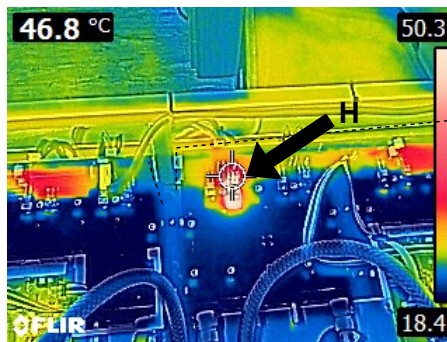

**Figure 9.** Thermal image of a Burn-in station undertaking burn-in of a Brick (**Left**). Thermal of an operational dummy load board VCC MOSFET affixed to a cooling plate (**Right**).

## 7. Summary and Outlook

The Tile-calorimeter Phase-II low-voltage power supply transformer-coupled buck converter project is in a mature state with delivery of the final production Bricks to CERN expected to occur in the first quarter of 2024. This date of delivery provides ample time for the Bricks to be assembled into the 256 LVPSs above ground before being installed during the Phase-II upgrade in 2026. The LVPS Brick is of an iterative design with the V8.6.0, the final version, being the result of various radiation, thermal, and reliability tests. These tests are required for the approval of the Bricks for use within the Phase-II TilCal and will ensure their reliable operation during HL-LHC collisions. The reliability of the Bricks is further improved by implementing a quality assurance testing procedure post manufacturing procedure that includes five distinct tests. Each test plays a specific role in the detection of latent/patent defects with the burn-in station forming the crux of this article. A single burn-in station, of which there will be a total of four, is tasked with the burn-in of eight Bricks at a temperature of 60 °C, a load of 5 A, and for a total duration of eight hours. The finalization of the burn-in procedure will take place after the application of the preliminary burn-in procedure to the pre-production batch of 104 Bricks which is expected to take place during the first quarter of 2023. The introduction of power/thermal cycling of the Bricks is currently being considered. The consistent and reliable application of the burn-in cycle requires the commissioning of all burn-in stations. The first burn-in station has undergone commissioning at the University of the Witwatersrand utilizing a now outdated V8.4.2 Brick. The test-station worked as designed but will require another series of commissioning tests with the final V8.6.0 Bricks. The same procedure is to be undertaken with the remaining test station at the University of the Witwatersrand and the two test stations at the University of Texas at Arlington in the coming months. Minor improvements, such as a power trip switch that is activated upon the opening of the burn-in station, are expected to be made in this time.

**Funding:** This research was funded in part by the National Research Foundation of South Africa, grant number 123017.

**Data Availability Statement:** Not applicable.

**Conflicts of Interest:** The author declares no conflict of interest.

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
