# Peer review of "A Burn-in Test Station for the ATLAS Phase-II Tile-Calorimeter Low-Voltage Power Supply Transformer-Coupled Buck Converters"

_instruments, doi:10.3390/instruments7010003_

Round 1
Reviewer 1 Report
Reliability is definitely a key requirement of all the components involved in the detector upgrades designed to operate at the HL-LHC. Low voltage power supplies are among such components.
I think that the paper is of general interest and should be published. I recommend the following minor modifications:
a) abstract: the sentence starting at line 8 it can be made more succinct. I do not understand why there is a need of such an elaborate description of the goal of burn-in, the screening of weak components that fail or show performance degradation after burn-in is a common feature of all burn-in systems. This applies also to the main text, where in line 92 there is another quite general description of possible deteriorations.
2) as this system is an evolutionary design, it would be nice to have a more elaborate motivation of the new features and relate them to the more demanding specifications of the HL-LHC.
3) line 90: start with Figure 2 shows and describe all the building blocks and functions.
4) line 95: the purpose of which... drop, this is kind of obvious. Can you comment on which weaknesses do you expect?
5) what was the rationale for the temperature and the duration of the tests?
6) paragraph starting with line 104: make more specific and justify the procedure (see 5). Is the system burn-in finalized on the basis of the experience with the previous versions? At this point section 5. should be incorporated and the details based on previous experience should be included.
Author Response
Greetings
Thank you for your detailed comments.
Please could you provide some clarification on comment 3?
From my understanding, would you like me to describe the block diagram in figure2?
My thanks.
Reviewer 2 Report
Dear Authors,
I found your article well written, interesting to read and with the
importance of the necessity for such a burn-in test station
appropriately highlighted.
Please find below a few minor suggestions, mainly to improve
readability for readers less familiar with the subject.
Best regards.
- Figure 2, please explain PRI and SEC in the text (e.g. line 76 for primary I suppose?). It will be helpful to the reader in section 6 in particular.
- line 134 mentions the Dummy-loads without explaining what it is. I suggest to add a sentence to explain the concept behind it, or break this sentence: e.g., if I understood correctly:
"A temperature control system....a key burn-in parameter. The bricks ultimately provide power to the detector modules, replaced by a so-called dummy load board (DLB) in the test setup. The temperature control system is also used to cool the DLB."
- Line 145 the symbol DLB is defined, whilst Figure 3 has "DL". Line 158 defines again DLB. I suggest to define it only once, for example like above, and use also the DLB acronym in Figure 3.
- Between L148-160 is quite hard to digest. I suggest to reorganise slightly and make shorter sentences, to explain the diagram of Figure 3 more linearly. Currently the text mentions MB then BIB then Brick then LIB then DLB (dummy load line 155), then Brick again and DLB again and then CP. My suggestion is to start from MB and then describe the path BIB to bricks on one side, then LIB to DL to Brick via CP from the other side. The PC and labview interface could be left for the end of this paragraph. Please also expand a little bit to explain op-amps and N-channel MOSFETs.
- L168 would it be appropriate to add a reference for "PVS60085MR HV power supply" ? Are those commercially available ?
- L177, I suggest to drop "for limiting the Burn-in....extrema" as this is already stated a couple of lines above, with two reasons expected.
- Section 6: I suggest to provide a short introduction to what tests were effectively performed, only one brick tested ? and maybe when also, to give a context to this paper in the future. It could also help the reader to refer again to Figure 2 at this point, for the mention of "primary-side" and MOSFETS (are those the FET driver in Fig 2 ? )
Author Response
Greetings
Thank you for your detailed comments.
I have attempted to address your comments in and amongst the comments of the other reviewers.
Please see the latest draft attached.
Best,
Ryan

Reviewer 3 Report
This paper serves to provide an overview of a Burn-in test station developed for performing extensive testing of the ATLAS Phase-II Tile Calorimeter low-voltage Power supply transformer-coupled buck converters. This upgrade and testing is necessary for the planned high-luminosity LHC upgrade project. The development of the test station along with preliminary results are presented.
General conceptual comments:
This paper describes a key testing capability which is important and essential for testing the ATLAS updated TileCal LVPS buck converters in order to ensure reliability during operations and minimize signal-loss regions in the detector during data taking. The procedure described and the test station details will serve as an important resource to the community. The LVPS brick and the description of the test station is presented well with enough details. However, the article as written doesn’t provide sufficient information to understand the finer details of the testing procedure and to evaluate the success of the testing station, and in particular, if the conclusions are strongly supporting the results presented. It could be due to where the current testing stands which is still preliminary but the lack of key details in important sections of the article makes it incomplete as currently presented. The article should be revised significantly for improved presentation in order to further consider for publication.
While the English language is overall well written in the article except for a few places, the grammar needs improvement. In particular, there are several places where commas were missing.
Introduction jumps right into the TileCal calorimeter without giving much context to non-experts about LHC or ATLAS and can be improved. I would recommend adding a paragraph to the introduction to briefly introduce LHC, ATLAS, where they are located including some of the high level physics goals. Also including details about how long ATLAS is running and what is the current status etc. will be useful
In section 1, provide some more details on the high luminosity LHC e.g. what luminosity, timeline etc. and make it comprehensive to understand what the resulting environment is expected to be and what aspect of it is necessitating these upgrades and more extensive testing procedures. If the ATLAS collaboration has certain design or operational requirements laid out, noting them in a tabular form will be useful to understand the overall context and will strengthen the motivation presented.
In section 2, it is mentioned that previous versions of these bricks were installed in ATLAS in 2007 and 2013, however, not much detail is provided as to how the performance was except for general statements like “generally functioned well” or in the case of V7.5.0, there was no mention of how they performed. More details in this regard are needed.
A 5-step testing procedure is listed in section 3 but each step is not much elaborated on except for the last two. For example, what does visual inspection and x-ray scan entail? What does initial testing mean? And for bricks failing during these first 3 steps, how does the testing procedure proceed etc. It is important to include these details for completeness and to understand the full procedure in testing these bricks. Related to this, are there any risks associated with the burn-in tests and related mitigations?
Section 4 should include details about the capacity of the burn-in station. How many bricks can be burn-in station test, only one or multiple at a time? What are the overall constraints w.r.t. The burn-in station’s capacity? Also, is this the only burn-in station available for ATLAS and does it have the capacity to test all of ATLAS’s Bricks for the full HL-LHC upgrades? Or, is this testing also distributed to other stations/centers? Some global context on this will be a useful detail to include.
Section 6 which I consider as a key section is extremely brief and will need more details such as how many bricks were tested and for how long, how many of those passed the tests, how many failed, what is the overall % of “weak” bricks in the batch etc. Figure 4 (right) is good to show an example of a successful burn-in test but I think other plots that capture the overall results of the full batch of bricks tested are important. For example, a plot of the overall distribution of the stable temperature achieved for various bricks; for bricks failing the test, is there a pattern in terms of temperature of load points where they are failing? -- including these details and elaborating the section will add more value to the paper.
Section 7 needs a better title than “Discussion”. I would suggest calling it “Summary and Outlook”. This summary can be more specific rather than saying “Favorable results”. Recapping the results in a quantitative way will be needed. More elaboration on the future plans will also be needed to make this section more complete, for example, what is the full batch of testing planned, timeline and how that aligns with the top-level schedule of ATLAS and the HL-LHC upgrades. Also, are there aspects of the burn-in station that will evolve in the future to increase capacity or general improvements planned?
Specific/minor comments:
Abstract: L8-13: run-on sentence. Suggest splitting that into two
L21: such “as”
Figure 1 resolution should be improved
Figure 1: Generated → generated
L36: add comma after “2029”
L59: remove “being”
L59: missing comma after “Bricks” at the beginning of the line
L61: missing comma after “upgrade”
L62: such “as”
L65: missing comma after “type”
L68: missing comma after “floating”
L73: provide reference for LT1681
L89: initiated → initiate
L94: missing comma after “phenomenon”
L100: on-detector → “on the detector”
L121: A physical image of the burn in station will be useful to visualize what is described in this section
L146-159: some acronyms are used without definition: VCCS, MOSFET, op-amps. While these maybe generally used acronyms, for clarity should be defined
L148: include reference for LABVIEW or BLA
L154: LIB provide → “LIBs provide” or “LIB provides”
L168: PVS60085MR needs a reference or a link. Not sure if HV is defined before
L176: The last part of the sentence is not well formed. “To allow” doesn’t sit right with the tense of the full sentence, maybe you need to say “allowing” or “enabling”
Figure 4 left: with the arrows, point out the main parts of the thermal image being shown. E.g where are the bricks, what are the other items being shown.
L197: stations → station’s
L223: this reference seems incomplete. Arxiv 2022 doesn’t specifically point to the reference. An arxiv number or a DOI link should be included.
Author Response
Greetings
Thank you for your detailed comments. I have attempted to address all of them but I must note that I have not provided any detail on the radiation testing of the Bricks. The motivation for this omission is that radiation testing does not fall under quality assurance testing but rather radiation hardness certification. A paper is currently being worked on that focuses on this topic in detail.
Best,
Ryan

Round 2
Reviewer 1 Report
I support publication of the current version
Author Response
Greetings
Our sincerest thanks for your assistance during the review process.
Best,
Ryan
Reviewer 3 Report
The authors have done a great job at incorporating all the feedback provided which has significantly improved the quality and value of the paper. The updated version reads well and has many important details included. I don’t have any major comments, only minor comments on the presentation and language which I list below. I recommend accepting the paper once the minor edits below are addressed.
L13-14: suggest rewriting the last sentence as “The commissioning of the burn-in station will be presented along with a summary and outlook of the project.”
L24, 25, 31: languages such as “Run 1”, Run 2”, “Run-3” are used.
-
These terms by themselves do not mean anything to outside readers except internally to ATLAS. I also noticed that these terms were not used in the rest of the document so I suggest removing this jargon and only mention the periods corresponding to these runs. Also, the way the sentence in L25 is written is confusing. I suggest rewriting it as: “...has progressively increased to 13 TeV in <insert month/year> and later to 13.6 TeV in July of 2022.”
-
If the author prefers to keep the jargon then I suggest they find a way to define them. E.g. “...has progressively increased to 13 TeV in <insert month/year> and later to 13.6 TeV in July of 2022. The 13 TeV (13.6 TeV) data taking period is referred to as Run 2 (Run 3) data.” Also note that consistent terms were not used for this on L24/25 and L31 (“Run x” vs “RUN-x”) . Adopt one consistent terminology if you choose to keep this
L34: missing comma after “This”
L36: “Its sub-detectors” → The sub-detectors of ATLAS”
Figure 1: the labels used on the figure use “LAr” but this is not defined anywhere in the caption or the paper. I would suggest remaking the labels to say “Liquid argon” or define it in the caption or main text where appropriate
L59: remove “the”
L62: remove “contained”
L73: remove “the” towards the end of the line
L77: remove “the” before “utmost”
L77-79: suggest rewriting the sentence as: “..assurance procedure as illustrated in Figure 4 has been developed which is applied to every Brick post-production.”
L83: Add “version” before “to be”
L84: “within” → “in”
L87: “Bricks the V7 Bricks” → “Bricks, the V7 Bricks”
L87: End the first part of the line at “designed” and start a new sentence: “...were designed. The V5.7.0 Bricks were installed in TileCal…”
L90-91: rewrite as “...per year over a period of 3 years since their installation in 2013.”
L99: Add “in this paper” after “detail”
L100: “acceptable” → “justified”
L110, 111: instead of “LVPSs”, say “LVPS modules”
L113: having → have
L123, 246: I noticed you forward reference figures which come much later in the article. For example on L123, you refer to figure 9 that is on page 10 of the article. Same on L246. Typically, figures are referenced sequentially in the order they appear. This is not a big deal but you can consider adjusting if possible.
I really like Figure 4, it is a nice one!
L161: weird space after “testing” before the period. Remove it
Figure 5 caption: What is “though-hole”?
“(1)” and “(2)” are used after MOSFET and diode – what are the numbers supposed to mean? It was unclear.
L191, 196: the opening single quote for “weak” is not well formed
L200: considered → described
L225: for “PVS60085MR” can you include a reference if available?
L262: “to within” → “to be within”
L273: stations → station’s
L274: remove “that”
L279: comma after “2023”
L290: “in” missing before “Figure 7”
L316: station → station’s
L339: Tilcal → TileCal
L341-343 is detailed for a summary. I would merge with the previous sentence and rewrite as: “post manufacturing which includes five distinct steps.” and remove the rest
L343: occupies → plays
L349: “place” missing after “take”
L355: where you say “three test stations” note locations of the stations for completeness
Author Response
Greetings
Thank you again for your detailed comments all of which have been addressed in the latest revision.
Best,
Ryan